# Generative Active Learning for Image Synthesis Personalization

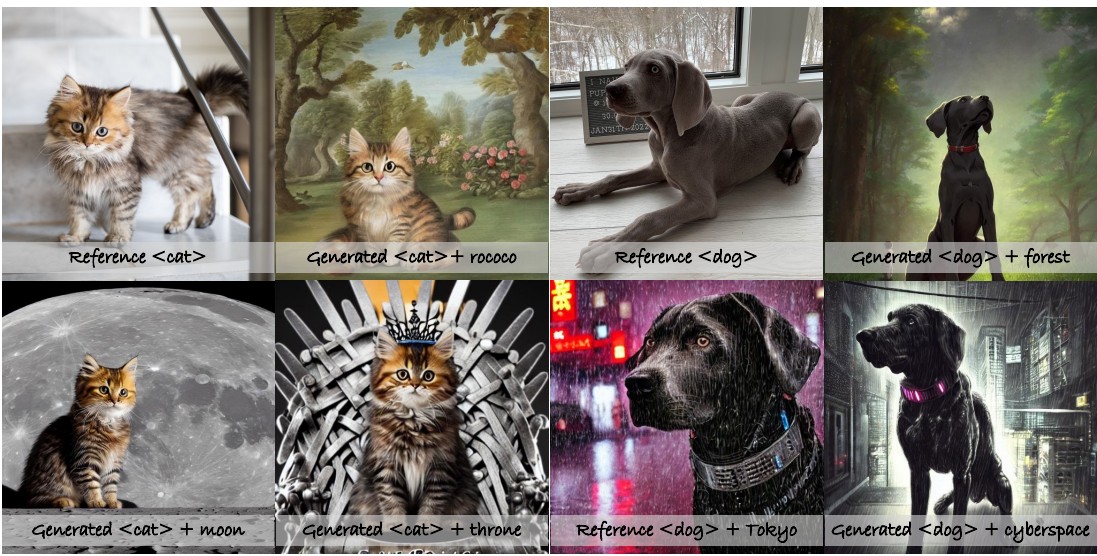

Figure 1: Given a few images of the subject of interest, the proposed method is capable of generating diverse personalized images in different contexts, such as moon, throne, rococo, etc.

## ABSTRACT

This paper presents a pilot study that explores the application of active learning, traditionally studied in the context of discriminative models, to generative models. We specifically focus on image synthesis personalization tasks. The primary challenge in conducting active learning on generative models lies in the open-ended nature of querying, which differs from the closed form of querying in discriminative models that typically target a single concept. We introduce the concept of anchor directions to transform the querying process into a semi-open problem. We propose a direction-based uncertainty sampling strategy to enable generative active learning and tackle the exploitation-exploration dilemma. Extensive experiments are conducted to validate the effectiveness of our approach, demonstrating that an open-source model can achieve superior performance compared to closed-source models developed by large companies, such as Google's StyleDrop. The source code is available at https://github.com/(open_upon_acceptance).

## CCS CONCEPTS

• **Computing methodologies** → **Artificial intelligence**.

## KEYWORDS

Generative Active Learning, Image Synthesis, Personalization

**ACM Reference Format:**
Anonymous Author(s). 2024. Generative Active Learning for Image Synthesis Personalization. In *Proceedings of ACM Multimedia 2024 (ACM MM)*. ACM, New York, NY, USA, 18 pages. https://doi.org/XXXXXXX.XXXXXXX

## 1 INTRODUCTION

Recently, generative models, such as large language models (e.g., ChatGPT [17], Llama [32]) and image generation models DALL•E [20], Stable Diffusion [22]), have demonstrated impressive capabilities in producing compelling and diverse results. The key to the success lies in the availability of high-quality training samples on an incredibly large scale. In addition to the real-world datasets that are expensive to collect, numerous studies [2, 16] have demonstrated that incorporating synthetic samples can effectively improve the capability and generalization of models. However, as the number of generated samples can be extensive and of varying quality, a crucial question arises: *how can we select the most informative samples with minimal cost for training?* This issue has been extensively discussed in the field of active learning, which attempts to maximize a model's performance while annotating the fewest samples [31]. However, traditional active learning approaches primarily focus on improving discriminative models. The application of active learning in generative models, particularly in utilizing synthetic samples to enhance model performance, remains an open and challenging research area.

In this paper, we present a pilot study on the application of active learning in generative models, specifically focusing on the image

synthesis personalization (ISP) [18]. ISP is a representative family of generative tasks that requires the cost-effective selection of synthetic data for training. The learning objective of ISP is to model the user's "subject of interest" (SoI) based on a limited number of reference images and generate new images that feature the SoI. For instance, in the case of learning from a few images of the user's pet cat, as illustrated in Figure 1, the trained model should be capable of generating diverse scenes with the cat, such as the cat on the moon or sitting on the Iron Throne, depending on the given prompt [23]. Similarly, when the SoI revolves around a specific style, like Van Gogh paintings or the user's own artwork, the ISP model should be able to adopt that style and generate new images with the same artistic characteristics [28]. Given the highly specific nature of personal interests, the availability of reference images is often limited. Therefore, selecting good samples from the newly generated images to augment the reference set has proven to be a more practical approach [28]. This can be done in an iterative manner, which aligns well with the framework of active learning.

While the idea of bringing active learning from discriminative models to generative models holds promise, it also presents several challenges. One key challenge is the causal loop in the querying strategy design. In discriminative active learning (DAL), informative samples are selected and queried from a closed set of unlabeled data, typically for tasks like recognizing predefined simple concepts (e.g., dog). This closed-set nature makes it feasible to design strategies that compare the information carried by different unlabeled samples (e.g., entropy in uncertainty sampling) so as to prioritize directions in the feature space for querying. In contrast, generative active learning (GAL) faces a scenario where the querying is open to all directions, because the user may combine the SoI with all possible prompts, which can carry much more complex and undetermined semantics. This openness makes the sample-evaluation-based DAL querying strategies infeasible in GAL. This is because generated samples are not readily available unless prompts are given, and it is not easy to design prompts before determining the directions to query. This creates a typical causal loop, making it challenging to establish a clear sequence of actions between determining what to generate and knowing which directions to query.

In this paper, we tackle this challenge by transforming the open querying problem into a semi-open one. Our approach involves collecting prompts to create a pool of querying intentions. The prompt embeddings serve as anchors in the target space, indicating the candidate directions to query and explore. During each iteration of the GAL process, we generate samples using these prompts for evaluation. This semi-open scheme strikes a balance by constraining the candidate directions for querying while allowing enough freedom to explore the target space through the generation of samples. Although this approach provides access to generated samples, the sample-based evaluation commonly used in DAL cannot be directly applied to GAL due to the fundamental differences between discriminative and generative models. Discriminative models learn a single distribution to distinguish simple semantics (e.g., dog), resulting in semantically consistent information carried by positive (negative) samples [21]. However, generative models focus on generalizing to various mixed semantics (e.g., to generate images not only of the user's pet dog in a forest but also of the dog on Tokyo street)

[20]. Consequently, generative models need to handle multiple sub-distributions, each modeling a specific combination of semantics. The information carried by samples from different sub-distributions are not consistent, rendering sample-based evaluation infeasible. To address this issue, we propose a distribution-based querying strategy that adapts the classical Uncertainty Sampling [31, 35] to the new generative scenario. It considers the distributional aspects of generative models and provides a more suitable framework for querying and evaluating samples in GAL.

Another challenge is the exploitation-exploration dilemma [37]. In DAL, the collected samples from different iterations are accumulated for training, and the learned distribution or decision boundary may gradually shift from the samples collected in the early iterations. This is generally not a problem as long as it benefits the classifier's performance. In contrast, in GAL, the fidelity to the reference images is of great importance which pushes the generated samples towards the references. Additionally, samples generated in the early iterations have been shown to have a higher likelihood of fulfilling the fidelity criteria compared to later iterations, and thus should be exploited as new references with greater attention. However, the generated samples cannot be too close to the references, otherwise, this causes over-fitting. Meanwhile, the generated samples need to be generalized to a certain target direction indicated by corresponding prompt, which attracts them to move toward the target direction against the references. The GAL process needs to learn how to navigate this balance between adhering to the references and exploring new directions. We propose a balancing scheme that evaluates the importance of references, thereby allowing us to weigh the contributions of different iterations.

The contribution of this paper can be summarized as: 1) A pilot work to discuss the application of active learning in generative models; 2) A distribution-based querying strategy for personalized image synthesis; and 3) A strategy to balance the exploitation and exploration in GAL.

## 2 RELATED WORK

### 2.1 Active Learning

Active learning is a subfield of machine learning, which aims to find an optimal querying strategy to maximize model performance with the fewest labeling cost. The most common strategies include uncertainty sampling [31, 35], query by committee [26], and representation-based sampling [8, 25], etc. The rationale behind is to provide the most valuable samples to learn a better decision boundary. However, acquiring real-world datasets still poses challenges in certain scenarios, such as few-shot learning. To address this issue, the use of generative networks for data augmentation has been investigated. For example, GAAL [43] first introduced GAN [9] to generate training samples. However, this random generation does not guarantee more informative samples compared to the original dataset. In contrast, BGADL [33] jointly trained a generative network and a classifier so as to generate samples in disagreement regions [31]. Subsequent approaches, such as VAAL [27] and TA-VAAL [12] employed adversarial training for data augmentation to improve the feature representation. It is important to note that while these works have explored the use of generative models, their primary focus is on improving the discriminative model's ability.

---

**Algorithm 1** Generative Active Learning

---

**Input:** anchor embedding set $\mathcal{A}$, reference images $\mathbf{x}$, subject of interest $\mathbf{e}^*$, non-SoI $\bar{\mathbf{e}}^*$, pre-trained model $f_\theta$, number of synthetic samples per prompt $m$
Initialize training set $\mathbf{T} = \{(\mathbf{x}, \mathbf{e}^* \oplus \bar{\mathbf{e}}^*)\}$.
**repeat**
    Fine-tune $f_\theta$ on $\mathbf{T}$
    **for** $\mathbf{a}_i$ in $\mathcal{A}$ **do**
        **for** $j = 1$ **to** $m$ **do**
            Generate image $\mathbf{I}_{ij}$ on $\mathbf{a}_i$ by $f_\theta$
            Verify whether $\mathbf{I}_{ij}$ is overfitted by Equation 8
        **end for**
        Calculate $\Omega(\mathbf{a}_i)$ according to Equation 6
    **end for**
    Update $\mathbf{T}$ with top-$k$ anchor embeddings
    Update openness score according to Equation 9
**until** Stopping criterion is met according to Equation 10

---

## 2.2 Personalized Content Generation

Text-to-image synthesis has earned significant attention for its potential applications in content creation, virtual reality, and computer graphics. Impressive works such as DALL•E [20], Stable Diffusion [22], Imagen [24], have shown immense potential to generate compelling and diverse images. As an application of image generation, personalized image synthesis offers user an opportunity to create customized object or style that is difficult to generate using pre-trained models. To accomplish content personalization, some studies [5, 14, 38, 39] have concentrated on training a unified model capable of personalizing any input image. However, these approaches struggle to perform satisfactory fidelity with the references. In contrast, other research studies [1, 4, 13, 23] enhance subject appearance preservation by adopting fine-tuning approach on pre-trained models for each reference group. In particular, Textual Inversion [6] aims to find an optimal token embedding to reconstruct the training images without additional regularization samples. DreamBooth [23] retrains the entire diffusion model and incorporates a prepared regularization dataset to alleviate the overfitting problem. Following this training framework and regularization approach, other works focus on enhancing different aspects of personalized image synthesis, like training acceleration [13] and multiple concepts composition [15, 30, 41]. As for expanding training samples, SVDiff [10] applies image stitching techniques, but does not explore the use of generated samples. In summary, although additional training samples are adopted in the training process, no generated samples are involved in these studies.

## 2.3 Personalized Style Generation

Style generation is one of the notable advancements in the field of image synthesis. Style transfer [7, 36, 42] aims to transform the visual style of a given image to another input image while preserving its contents. However, these methods do not offer the chance to generate images based on text prompts. Meanwhile, another line of research focuses on personalized style generation, which aims to reverse visual styles on textual descriptions. A recent study, StyleDrop [28], introduces a parameter-efficient fine-tuning method and

an iterative training framework with feedback to facilitate style recreation. Specifically, preset prompts are used to generate images and these images are then subject to user filtering, where users will identify high-quality images that can be used for further training. While this approach leverages human feedback to enhance model performance, the need for human inspection and the equal weighting of selected samples pose limitations. In this paper, we propose methods that effectively alleviate the burden on human resources through active learning and reduce selection bias by balancing the importance of synthetic and real samples.

## 3 METHOD

In this section, we introduce our implementation of generative active learning for image synthesis personalization. The algorithm, along with its pseudo-code, is depicted in Algorithm 1.

### 3.1 Preliminaries for Image Synthesis Personalization

The current state-of-the-art methods for Image Synthesis Personalization (ISP) are all based on diffusion models [11, 29]. What sets diffusion models apart is their "generate-by-denoise" approach. During training, a text-image pair is used, and the process begins by iteratively adding noise to the image $\mathbf{x}$ according to the Markov chain, resulting in a noisy image $\mathbf{x}_t$. The noisy image is then combined with the text embedding $\mathbf{e}$ to create a new noisy image embedded with the text semantics, denoted as $\mathbf{x}_t \circ \mathbf{e}$. Learning then proceeds to denoise this image and reconstruct the original image $\mathbf{x}$, which is represented as

$$\hat{\mathbf{x}} = f_\theta(\mathbf{x}_t \circ \mathbf{e}) \tag{1}$$

The objective is to minimize the reconstruction loss

$$L_{rec} = \mathbb{E}\left[w_t \|\hat{\mathbf{x}} - \mathbf{x}\|_2^2\right] \tag{2}$$

where $w_t$ is a time-dependent weight. During the inference, the prompt embedding $\tilde{\mathbf{e}}$ is then fused with a random noise $\epsilon$ to generate the image $\tilde{\mathbf{x}} = f_\theta(\epsilon \circ \tilde{\mathbf{e}})$ that aligns with the semantics of interest.

To perform an ISP process, a pre-trained model $f_\theta$ is typically fine-tuned using reference images that contain the Subject of Interest (SoI). A pseudo text word $S^*$ is utilized to represent the SoI and is incorporated into simple sentences, such as "a photo of $S^*$," as a reference prompt. The training process involves updating the parameters of the model $f_\theta$ to establish the association between the visual appearance of the SoI (indicated by given reference images $\mathbf{I}_r$) and its corresponding semantic embedding $\mathbf{e}^*$. After the fine-tuning, new images of SoI can be generated with prompts like "$S^*$ running on the street with a dog" or "$S^*$ ridding a house on the Golden Bridge" if the SoI is an object. In case the SoI is a specific style, new images can be generated using prompts like "a drawing of New York City with style $S^*$" or "a teddy bear of style $S^*$".

### 3.2 Direction-based Uncertainty Sampling

It is evident that a limited number of reference images for the SoI is insufficient to ensure the fine-tuned model's generalizability to a broader range of semantics. We need to generate new samples to augment the references, which requires prompts to determine the direction to query. However, the querying remains open to all

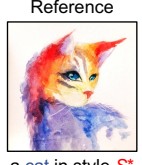 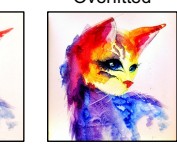 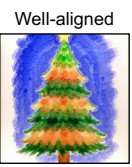

Reference | Overfitted | Well-aligned

a cat in style $S^*$
(non-SoI: cat
SoI: $S^*$)

a tree in style $S^*$
(overfitted to non-SoI)

a tree in style $S^*$
(non-SoI excluded
and SoI fitted)

**Figure 2: Overfitted and well-aligned generations. The model has to exclude the non-SoI for successful generations.**

directions since users may combine the pseudo text word $S^*$ with various unseen concepts in future prompts. To address this, we transform the problem into a semi-open one by incorporating the SoI with a set of predefined concepts (e.g., cat and table) that can be gathered from existing benchmarks. These concepts serve as anchors in the target space, with each anchor representing a specific direction for querying when combined with the SoI to form anchor prompts (e.g., "$S^*$ with a dog"). The model's ability to generate high-quality samples for these anchor prompts determines its level of generalization. While the anchor directions are predetermined, the querying process remains open due to the introduction of random noise $\epsilon$, which leads to variations in the generated images for the same prompt. To ease the discussion, let us denote the anchor embedding set as $\mathcal{A} = \{\mathbf{a}_i\}, i \in \mathbb{N}$ and the set of embeddings of anchor prompts or directions to query can be denoted as

$$\mathbf{e}^* \oplus \mathcal{A} = \{\mathbf{e}^* \oplus \mathbf{a}_i\}, i \in \mathbb{N}. \tag{3}$$

where $\oplus$ is a model-dependent operator, which is typically implemented by directly inputting the embeddings as a sequence.

In each iteration of the generative active learning (GAL), we generate $m$ samples for each anchor prompt. We need to initiate the next round of GAL by selecting informative ones from the generated samples as new references. However, as discussed, conventional sample-based querying is infeasible in GAL, because evaluating performance on individual samples lacks of global perspective to measure the model's generalizability. Additionally, relying solely on generalizability to build a metric is challenging because higher generalizability may indicate well-explored directions, where samples would not provide novel information for improving the model. This is similar to the situation in Discriminative Active Learning (DAL), where including samples from well-classified locations does not contribute to performance improvement and instead hinders exploration. A popular solution is Uncertainty Sampling [31], which selects samples from areas where the model exhibits uncertainty. In the context of GAL, we can adapt this idea to identify directions where the quality of model-generated samples lies between well-generalized and overfitted. Let $\mathbf{I}_{ij}, j \in [1, m]$ denote an image generated for the $i^{th}$ anchor direction $\mathbf{a}_i$ as

$$\mathbf{I}_{ij} = f_\theta \left( \epsilon \circ (\mathbf{e}^* \oplus \mathbf{a}_i) \right) \tag{4}$$

and there is an oracle function to verify whether $\mathbf{I}_{ij}$ is overfitted as

$$\Phi(\mathbf{I}_{ij}) \in \{0, 1\}, \tag{5}$$

we can implement a direction-based uncertainty sampling for GAL by measuring the entropy on the portions of overfitted (non-overfitted)

samples as

$$\Omega(\mathbf{a}_i) = - \left[ (1 - \beta_i) \log(1 - \beta_i) + \beta_i \log \beta_i \right] \tag{6}$$

$$\beta_i = \frac{\sum_{j=1}^m \Phi(\mathbf{I}_{ij})}{m}. \tag{7}$$

In DAL, the learning employs human annotators as oracles. However, due to the computational expense of current diffusion models, it becomes impractical for human annotators to wait for the results of each iteration, resulting in significant delays. Hiring human annotators as oracles can be extremely costly, which might be one of the reasons why successful ISP models using generated results as argumentation are predominantly developed by large companies like Google [5, 28], who can afford such expenses. In our study, we found the oracle function $\Phi(\mathbf{I}_{ij})$ can be estimated by evaluating the generated image $\mathbf{I}_{ij}$'s fidelity to both the anchor direction and irrelevant semantics in the reference prompt. This observation stems from the fact that the reference prompt consists of two components: the SoI and non-SoI semantics. Most previous studies focus on the fidelity to the SoI semantics, while the non-SoI semantics are not fully leveraged. These non-SoI semantics can be considered distractor semantics that the generated images should avoid, similar to negative labels in discriminative models. One such example can be found in Figure 2, in which the SoI is the drawing style while the non-SoI is the concept cat. The overfitted samples are those failed to disentangle the cat from the generation. Therefore, we propose a straightforward metric to simulate the oracle function. Let $\bar{\mathbf{e}}^*$ denote the non-SoI embedding, the function is written as

$$\Phi(\mathbf{I}_{ij}) = \begin{cases} 1, & sim(\mathbf{I}_{ij}, \mathbf{a}_i) \leq sim(\mathbf{I}_{ij}, \bar{\mathbf{e}}^*) \\ 0, & sim(\mathbf{I}_{ij}, \mathbf{a}_i) > sim(\mathbf{I}_{ij}, \bar{\mathbf{e}}^*) \end{cases} \tag{8}$$

where the $sim()$ is a fidelity metric of an image to a semantics. In this study, we simply adopt the CLIP similarity [19].

With all necessary components built, the querying can then be conducted by evaluating all the anchor directions and selecting the ones with top-$k$ uncertainty scores (using Equation 6). For each direction, we choose the generated image with the highest $sim(\mathbf{I}_{ij}, \mathbf{a}_i)$ score (indication of the faithfulness to the direction) as a new reference image.

## 3.3 Balancing the Exploitation and Exploration

As aforementioned, in the progression of GAL iterations, we need to keep the knowledge learned at past rounds while encouraging the model to explore. This introduces an exploitation-exploration dilemma [37]. To address this challenge, we propose evaluating the openness of the model at each round, using it as an indicator of the expected contribution of the novel information introduced in that round. Given that the novel information is encapsulated within the newly included reference images, we can utilize this indicator as a weight to regulate their impact on the learning process in the subsequent round. This encourages the exploration when the expected contribution is high, otherwise encourages the exploitation.

To assess the openness of a round, we can utilize the uncertainty score previously computed by Equation 6. Our rationale is that as the model explores more directions, its level of openness increases. Hence, the openness score for the round can be estimated

**Table 1: The performance of different GAL strategies including random selection (Random), human feedback (Human), direction-based uncertainty sampling (Uncertainty), direction-based uncertainty sampling with balance scheme (Uncertainty + Balance), human feedback with balance scheme (Human + Balance).**

| Models | Object | | | Style | | |
|---|---|---|---|---|---|---|
| | TXT-ALN ↑ | IMG-ALN ↑ | OVF ↓ | TXT-ALN ↑ | IMG-ALN ↑ | OVF ↓ |
| Baseline (DreamBooth) | 0.298 | **0.796** | 0.363 | 0.318 | **0.694** | 0.171 |
| Random | 0.285 | 0.714 | 0.391 | 0.247 | 0.620 | 0.327 |
| Human | 0.297 | 0.721 | 0.331 | 0.272 | 0.622 | 0.212 |
| Uncertainty (ours) | 0.305 | 0.755 | 0.268 | 0.286 | 0.628 | 0.110 |
| Uncertainty + Balance (ours) | **0.309** | 0.771 | 0.268 | 0.337 | 0.669 | 0.058 |
| Human + Balance (Oracle + ours) | 0.307 | 0.772 | **0.254** | **0.342** | 0.650 | **0.023** |

by calculating

$$\Delta(f_\theta) = \frac{\lambda}{|\mathcal{A}|} \sum_{i=1}^{|\mathcal{A}|} \Omega(\mathbf{a}_i) \qquad (9)$$

where $\lambda$ is a learning rate. This can be used to weight the newly include reference images to control their degrees of influence to the loss $L_{rec}$ (Equation 2).

An additional outcome of Equation 9 is its potential to establish an adaptive stopping criterion for GAL learning, in contrast to the fixed number of iterations often set in DAL. The concept behind this approach is to halt the learning process when there are fewer directions left to explore than anticipated. The stopping criteria is then simply written as

$$\left| \{ \Omega(\mathbf{a}_i) \mid \Omega(\mathbf{a}_i) > 0, \mathbf{a}_i \in \mathcal{A} \} \right| < k. \qquad (10)$$

## 4 EXPERIMENTS

**Datasets.** To evaluate the performance of active learning in ISP, we conduct experiments on two most representative tasks, style- and object-driven personalization.

For style-driven ISP, we adopt the evaluation dataset used in the StyleDrop [28]. This dataset comprises various styles, such as watercolor painting, oil painting, 3D rendering, and cartoon illustration. 190 basic text prompts sourced from the Parti prompts dataset [40] are used to generate images, yielding 36,480 images.

For object-driven ISP, we adopt almost all concepts that have been previously used in related studies [6, 13], comprising a total of 10 categories including animals, furniture, containers, houses, plants, and toys. We use the 20 prompts in [13], which cover a wide range of test scenarios. In total, this process generates 6,400 images for a complete training cycle.

**Evaluation Metrics.** We utilize three metrics: 1) *Text-alignment* (TXT-ALN) assesses how well the generated images align with the intended textual descriptions This can be implemented by calculating the similarity between the CLIP image feature and the text feature. 2) *Image-alignment* (IMG-ALN) measures the extent to which the generated images capture the content or style present in the reference images. This can be implemented by the CLIP feature similarity between reference images and generated images. 3) *Overfit* (OVF) evaluates the proportion of overfitting in the test samples based on Equation 8. Lower scores indicate better performance in terms of generalization and avoiding overfitting.

**Base Model.** DreamBooth [23] is a widely adopted method with promising generation results. Thus, we utilize DreamBooth as our baseline, with the first-round results derived directly from it without synthetic training data. For our proposed method, we set the values of $m$ and $\lambda$ to 10 and 0.005, respectively. The initial anchor directions comprise 18 prompts. We select top-3, along with their associated highest-fidelity images, to serve as additional training pairs. We provide more implementation details in the **Appendix**.

### 4.1 Does generative active learning work in ISP?

To evaluate the performance of different strategies, we compare our method with two commonly adopted querying strategies, including Random Sampling and Human Sampling. To be fair and efficient, we set a maximum number of rounds to 4 in all experiments. The initial round is based on original references without any synthetic data. The results are shown in Table 1. It is evident from the results that both Random and Human strategies do not necessarily enhance the baseline performance. Instead, these strategies show a degradation on style-personalization of 22.3% (14.5%), 10.7% (10.4%), and 91.2% (24.0%) on TXT-ALN, IMG-ALN, and OVF, respectively. The unexpected degradation observed in the Human strategy, which is often considered an oracle in DAL, confirms the fundamental distinction between discriminative and generative tasks: while human annotators can easily differentiate between positive and negative samples, evaluating generalizability is a more challenging aspect. Therefore, we integrate human annotators with our balancing scheme to create a run that combines their selections and fairly weighs them for improved learning. The results are shown as the last row in Table 1 which demonstrates an approaching optimal performance and thus can be used as an oracle. Spuriously, our proposed method (Uncertainty+Balance) achieves a comparable performance with the oracle run. This validates its effectiveness.

### 4.2 How does the uncertainty sampling work?

To gain deeper insights, we conduct a case study to observe the rationale behind the uncertainty sampling. Figure 3 shows the distribution of images generated by the anchor prompts. Within the feature space, multiple sub-distributions can be observed. One particular distribution is centered around the reference, consisting of poor-quality images that exhibit non-SoI of the reference, such as the failure cases illustrated in Figure 3. In contrast, images that align well are located far from the reference, forming smaller distributions that exclude non-SoI, like the successful samples of generated

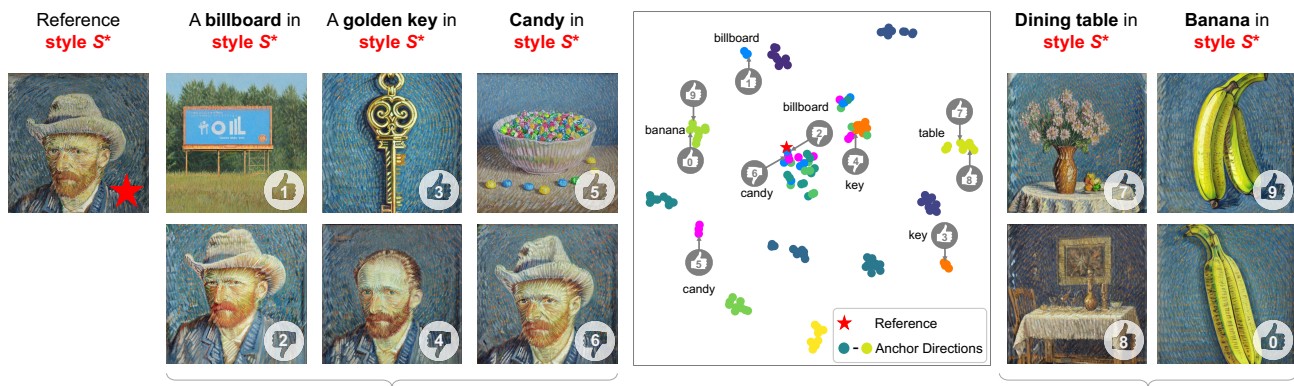

Groups exhibiting HIGH entropy with diverse sample distributions

Groups with LOW entropy and homogeneous distributions

**Figure 3: Examples of images generated by anchor prompts in round 2 with higher priority (left) and lower priority (right). Their CLIP image features are highlighted in the tSNE [34] space (middle). Poor-quality images that exhibit non-SoI are distributed near the reference, while high-quality images are located far from the reference.**

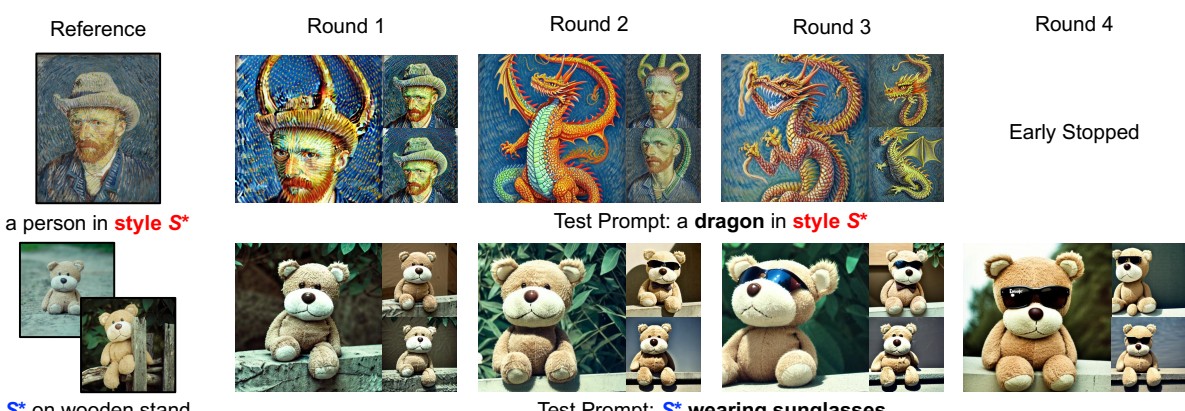

**Figure 4: Results of GAL over iterations. The images shown in the $1^{st}$ and $2^{nd}$ groups are for style- and object-driven ISP, respectively. The non-SoI and SoI are gradually disentangled and dragons or glasses are generated. Additional examples are available in the Appendix.**

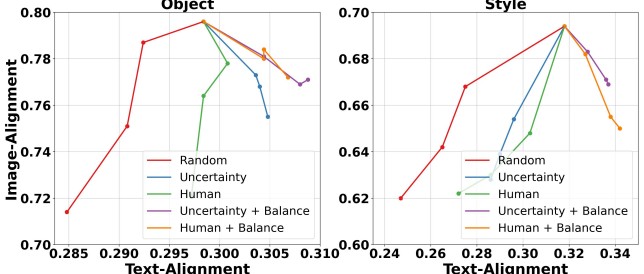

**Figure 5: The curves shown in the figure resemble clock arms extending from the baseline performance points. As these arms move in an anti-clockwise direction towards the top-right corners, better performance is observed.**

billboard, key, and candy images. Additionally, the distributions of good and bad samples across these three directions demonstrate

significant diversity, suggesting a limited ability to generalize along these directions. As a result, these directions are given higher priority for querying based on our uncertainty metric. On the other hand, distributions at the directions of table and banana are homogeneous. Consequently, these directions exhibit lower entropy and lower querying priority. This observation aligns with the rationale we presented earlier.

## 4.3 How does GAL progress over iterations?

To examine the progress of GAL over iterations, we present the performance in each round, as shown in Figure 5, and visualize the evolution through the cases in Figure 4. One notable observation is the dramatic and consistent decrease in performance of the Random strategy due to the inferior samples by random selection. After adopting a better querying strategy, the rate of decrease becomes much slower, and Uncertainty sampling begins to outperform the

**Table 2: Comparison with SOTA methods for object-driven ISP. Results marked with $^\dagger$ indicate our re-implementation using publicly available codebases.**

| Models | TXT-ALN | IMG-ALN | OVF |
|---|---|---|---|
| IP-Adapter$^\dagger$ | 0.270 | **0.858** | 0.734 |
| Textual Inversion$^\dagger$ | 0.277 | 0.778 | 0.441 |
| Custom Diffusion$^\dagger$ | 0.301 | 0.776 | 0.287 |
| DreamBooth | 0.298 | 0.796 | 0.363 |
| + Uncertainty + Balance (R2) | 0.304 | 0.781 | 0.300 |
| + Uncertainty + Balance (R3) | 0.308 | 0.769 | **0.248** |
| + Uncertainty + Balance (R4) | **0.309** | 0.771 | 0.268 |
| + Oracle + Bablance (R4) | 0.307 | 0.772 | 0.254 |

**Table 3: Comparison with SOTA methods for style-driven ISP. Results marked with $^\ddagger$ are obtained from [28].**

| Models | TXT-ALN | IMG-ALN | OVF |
|---|---|---|---|
| Imagen$^\ddagger$ | 0.337 | 0.569 | - |
| DB on Imagen$^\ddagger$ | 0.335 | 0.644 | - |
| Muse$^\ddagger$ | 0.323 | 0.556 | - |
| StyleDrop$^\ddagger$ | 0.313 | **0.705** | - |
| StyleDrop-Random$^\ddagger$ | 0.316 | 0.678 | - |
| StyleDrop-CF$^\ddagger$ | 0.329 | 0.673 | - |
| StyleDrop-HF$^\ddagger$ | 0.322 | 0.694 | - |
| DreamBooth | 0.318 | 0.694 | 0.171 |
| + Uncertainty + Balance (R2) | 0.328 | 0.683 | 0.097 |
| + Uncertainty + Balance (R3) | 0.336 | 0.671 | 0.059 |
| + Uncertainty + Balance (R4) | 0.337 | 0.669 | 0.058 |
| + Oracle + Balance (R4) | **0.342** | 0.650 | **0.023** |

baseline on TXT-ALN for object-driven personalization, which suggests the effectiveness of valuable samples in enhancing generative models. The best overall progress is achieved through the combined strategies of Uncertainty sampling and the balancing scheme. We can find that TXT-ALN consistently improves and reaches its highest alignment in round 4, while IMG-ALN remains within a reasonable range. This trend is evident in Figure 4, where the non-SoI semantics gradually disappear, and the number of successful generations of glasses placed on $S^*$ or dragon in style $S^*$ increases. Meanwhile, the SoI is maintained throughout the iteration rounds. These results indicate a progressive improvement by GAL as the iterations proceed. Additional comprehensive examples are available in the **Appendix**.

### 4.4 Comparison with SOTA methods

For object driven-personalization, we compare 4 popular state-of-the-art (SOTA) methods including Textual Inversion [6], Custom Diffusion [13], DreamBooth [23], IP-Adapter [39]. The results are shown in Table 2. Compared to IP-Adapter, Textual Inversion, and Custom Diffusion, our method demonstrates significant improvements on TXT-ALN and OVF throughout almost all rounds, achieving 14.4%(63.5%), 11.6%(39.2%), and 2.7%(6.6%) on TXT-ALN (OVF) in terms of round 4. Since the non-SoI semantics dominate the outputs of the other approaches, our method exhibits a slight decrease on IMG-ALN. Figure 6 provides visual evidence of our method's

**Table 4: The percentage of user preference on our proposed method (Uncertainty + Balance) compared to Round 1 (DreamBooth) and Oracle feedback (Human + Balance).**

| | Object | | Style | |
|---|---|---|---|---|
| | TXT-ALN | IMG-ALN | TXT-ALN | IMG-ALN |
| Ours vs. Round 1 | 60.4 % | 32.5 % | 77.8% | 59.8% |
| Ours vs. Oracle | 53.8 % | 46.2 % | 47.0% | 67.8% |

superior text and object fidelity. The success in higher text fidelity can be observed in the accurate placement of the cat statue in the Grand Canyon and the realistic interaction between the marigold flowers and the teapot. Furthermore, our method enhances object fidelity by accurately reconstructing only one spout and better preserving the color of the cat statue.

For style-driven personalization, we conduct a comparison between four variations of StyleDrop [28]: base model, random feedback (StyleDrop-Random), clip-based feedback (StyleDrop-CF), and human feedback (StyleDrop-HF). Additionally, we include the results of DreamBooth on Imagen [24] as well as other pre-trained models like Imagen and Muse [3], as reported by [28]. It is clear that our method significantly outperforms the pre-trained models and achieves superior performance in terms of 4.7% and 2.4% on TXT-ALN compared to the dedicated human feedback and clip-based feedback of StyleDrop. It is worth noting that the closed-source StyleDrop is built on a more powerful backbone, Muse, compared to Stable Diffusion. This indicates that the open-sourced ISP models are able to achieve better performance with GAL.

### 4.5 User Study

We conduct a user study involving two comparison tasks. Participants are presented with reference images and a text prompt, and are asked to choose the more faithful result in terms of object/style and text fidelity. This process yields a total of 4800 responses from 8 participants. The results are shown in Table 4. It is clear that our method significantly improves the text alignment, with particularly notable gains in style-driven ISP where both text and style fidelity surpass round 1. This indicates the superior performance of GAL when users can only provide fewer samples. By comparing the oracle feedback, our automatic uncertainty sampling strategy performs comparable results. Notably, a majority of users prefer our style renderings rather than those trained from human selection. This further validates the effectiveness of our method. More details are available in **Appendix**.

### 4.6 Ablations

Figure 7 presents ablation studies on style-driven ISP to evaluate our model's sensitivity to various hyperparameters. Details on the ablations for object-driven ISP are provided in the **Appendix**.

**Learning Rate $\lambda$ on Openness.** Subfigures (a) depicts the effect of the learning rate $\lambda$, which controls the scale of the openness score in Equation 9. It is obvious that a relatively higher $\lambda$ does not exhibit promising results, particularly in scenarios with a single reference for style-driven ISP. Additionally, a $\lambda$ below 0.05 results in stable performance.

**Size of Anchor Set.** As illustrated in subfigures (b), increasing the size of the anchor embedding set enhances style fidelity but

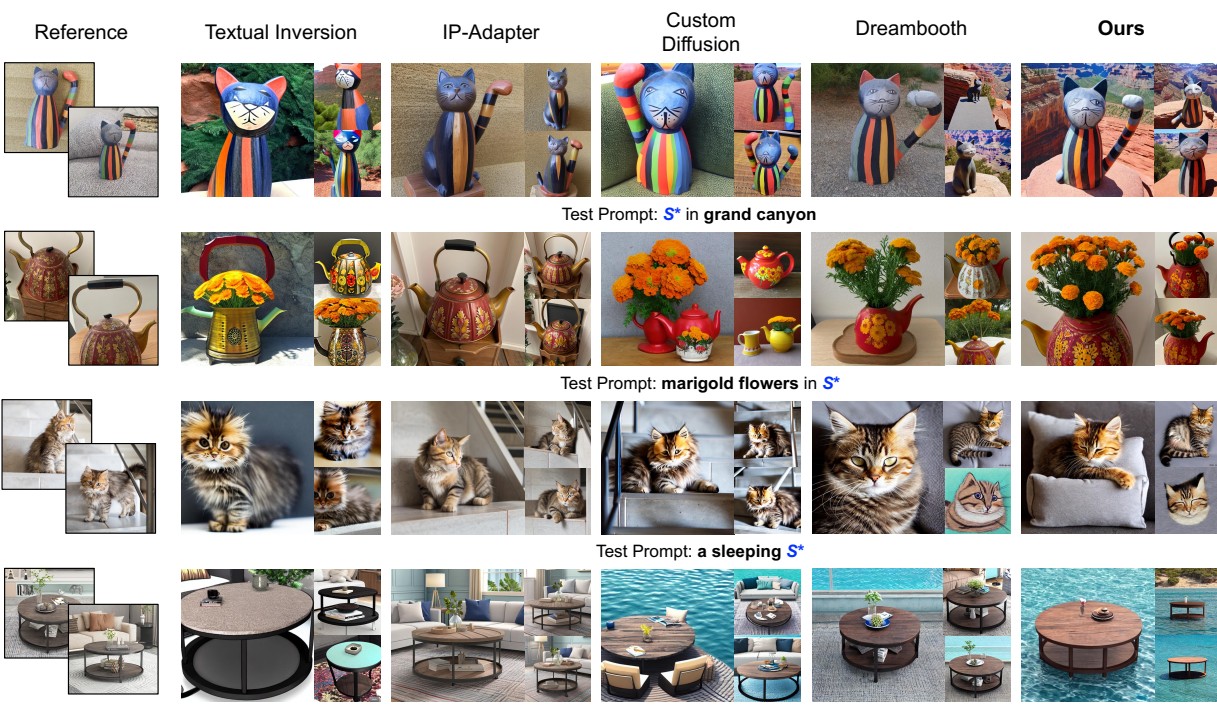

| Reference | Textual Inversion | IP-Adapter | Custom Diffusion | Dreambooth | **Ours** |

Test Prompt: **S\*** in **grand canyon**

Test Prompt: **marigold flowers** in **S\***

Test Prompt: **a sleeping S\***

Test Prompt: **S\* floating** in the **ocean**

**Figure 6: Qualitative comparison between our method and SOTA methods for personalized content generation. Our method produces text-aligned images compared with other methods. Additional comprehensive examples are available in the Appendix.**

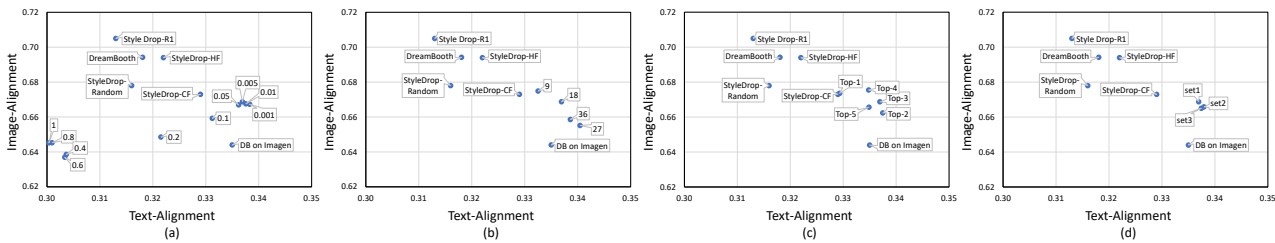

**Figure 7: Illustration of ablation experiments on style-driven ISP. (a) Variation in performance with the parameter $\lambda$. (b) Effects of different anchor set sizes. (c) Impact of selecting the top-$k$ prompts per iteration. (d) Results from varying the prompt composition within the anchor set.**

reduces text alignment. Conversely, a smaller anchor size exhibits the opposite effect. Therefore, we consider a moderate size of 18 as our default setting.

**Top-$k$ Anchor Prompts.** Because of the trade-off between IMG-ALN and TXT-ALN metrics, as shown in subfigures (c), there is no globally optimal top-$k$ setting. Consequently, we adopt the top-3 selection as a standard practice based on relative performance.

**Anchor Set Variability.** Finally, we change the prompts in the anchor set, forming 3 distinct sets, each differing by at least 50%. As shown in subfigures (d), the results reveal our model's robustness against variations in anchor prompts. This indicates the effectiveness of our uncertainty sampling method which selects the most constructive direction for model training.

## 5 CONCLUSION

This paper presents a pilot study that investigates the application of active learning to generative models, specifically focusing on image synthesis personalization tasks. To solve the open-ended nature of querying in generative active learning, this paper introduces anchor directions, transforming the querying process into a semi-open problem. An uncertainty sampling strategy is introduced to select informative directions, and a balance scheme is proposed to solve the exploitation-exploration dilemma. Through extensive experiments, the effectiveness of the approach is validated, indicating new possibilities for leveraging active learning techniques in the context of generative models.

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
