# OpenReview forum: "Generative Active Learning for Image Synthesis Personalization"
_acmmm.org/ACMMM/2024/Conference — MM2024 Poster_

### Official Review · Reviewer_18M8 · 2024-05-20

**Rating:** 5
**Confidence:** 3

**Summary:**

This paper applied active learning to Image Synthesis Personalization for the first time, where samples are limited and requires efficient utilization. Different from discriminative models, the generative models encounter challenges like open-ended querying and the diversity of generation outputs. To solve these problems, the authors transformed the open querying problem into a semi-open one by predefining a concept set as anchors for querying, and proposed a direction-based uncertainty sampling by automatically detect overfitted generations. Comprehensive and ablation experiments justified the effectiveness of the proposed active learning framework.

**Strengths:**

Strengths:

1.	The direction-based uncertainty sampling based on a set of predefined concept set is interesting, which fully utilize the limited data to approximately solve the open-ended querying problem.

2.	The proposed oracle function, which determines whether a generation is overfitted, is simple and automatic, and further leverages the non-SoI information to improve generalizability.

**Limitations:**

Weaknesses & Questions:

1.	Some notations are not cleared, for example, the k in Equation (10).

2.	The authors should improve the writings of section introduction, related work, and method. For example, from Para 3 to Para 5 in section Introduction, it is difficult to grasp the main challenge, idea or procedure of active learning in ISP. And for the end of section 2.1 and 2.2, the authors are expected to add some descriptions of the differences between the proposed work and related works since they have mentioned what related works focus on.

3.	In section 3, the authors only mentioned the Algorithm 1 at the beginning of this section, which makes it really hard to understand at first glimpse. Furthermore, why the training set T doesn’t contain the anchor embedding set A?

**Suitability:**

2

---

### Official Review · Reviewer_Tkqo · 2024-05-24

**Rating:** 4
**Confidence:** 3

**Summary:**

This paper explores the application of active learning (AL) to generative models (GM), particularly in the context of personalized image synthesis tasks. Traditionally, active learning has been studied extensively in the domain of discriminative models (DM), but this research extends it to generative models. This paper focus on the image synthesis personalization (ISP). To tackle the open-ended query issue and the exploration-exploitation dilemma in generative active learning, this paper introduces the concept of anchor direction and proposes a direction-based uncertainty sampling strategy. Extensive experiments validate the effectiveness of the proposed methods. It provides a new solution for the task of image synthesis personalization.

**Strengths:**

1. This paper extends active learning to generative models, addressing the unique challenges involved. Providing new methods and insights for personalized image synthesis tasks.
2. The paper proposes an uncertainty sampling strategy based on directions to select information-rich directions during the generative active learning process. Solve the open-ended nature of querying in generative active learning
3. Extensive experiments validate the effectiveness of the proposed methods

**Limitations:**

1. When considering evaluation criteria, is it overly simplistic to utilize CLIP for the direct assessment of image alignment? Could factors other than style have influenced the assessment？
2. Style-driven and object-driven approaches exhibit varying levels of granularity in terms of image similarity.  Would it be inappropriate to apply identical standards across both approaches?

**Suitability:**

2

---

### Official Review · Reviewer_6mPi · 2024-05-24

**Rating:** 4
**Confidence:** 3

**Summary:**

This paper explores the possibility of using active learning in the context of generative models. The authors choose to focus on the image synthesis personalization task. Since generative models are oriented towards open scenarios, the authors introduce anchor directions to transform the querying process from an open problem to a semi-open problem, and further propose a direction-based uncertainty sampling strategy and a trade-off between exploitation and exploration. Sufficient experiments demonstrate the effectiveness of the method.

**Strengths:**

+ Novelty. The authors explore the application of active learning in generative models. This is important for the further development of active learning, and it is also what I have always wanted to do.
+ Sound methods and sufficient experiments. The proposed method seems practical. Although the method is not complicated, it is sufficient as a pilot study. The authors conducted sufficient experiments to prove the effectiveness of the proposed method.

**Limitations:**

- Lack of in-depth analysis of scenarios for generative models. Since this is an attempt to apply active learning on generative models, the generative model scenario deserves a detailed analysis. On the one hand, this will help readers better understand the main challenges raised by the authors (such as the exploitation-exploration dilemma). On the other hand, this will provide an explanation for the rationality of the anchor directions proposed by the authors (which the authors have not yet explained).
- The presentation needs to be improved. Although the authors have spent a considerable amount of space to introduce the background and the proposed method, there are still some places that need further explanation (especially the background introduction of generative models and the explanation of the method details).

**Suitability:**

3

---

### Meta-Review · Area_Chair_TPBA · 2024-06-28

**Recommendation:** Accept (Poster)
**Confidence:** 4

**Metareview:**

It is a nice paper of exploring the application of active learning in generative models. The work is focused on the image synthesis personalization task, proposes a new method to solve the open-ended nature of querying in generative active learning, and achieves impressive results. All reviewers agree to accept it for publication.